

# Evaluation of tropospheric ozone and ozone precursors in simulations from the HTAPII and CCMI model intercomparisons - a focus on the Indian Subcontinent.

Zainab Q. Hakim[1], Scott Archer-Nicholls[1], Gufran Beig[6], Gerd A. Folberth[3], Kengo Sudo[4,5], Nathan Luke Abraham[1,2], Sachin Ghude[6], Daven Henze[7], Alexander T. Archibald[1,2,*]

[1]. Department of Chemistry, University of Cambridge, Cambridge, U.K.

[2]. National Centre for Atmospheric Science, U.K.

[3]. UK Met Office Hadley Centre, Exeter, UK

[4]. Graduate School of Environmental Studies, Nagoya University, Nagoya, Japan

[5]. Japan Agency for Marine-Earth Science and Technology (JAMSTEC), Yokohama, Japan.

[6]. Indian Institute of Tropical Meteorology, Pune, India.

[7]. Department of Mechanical Engineering, University of Colorado, Boulder, CO, USA.

*Correspondence to: A. T. Archibald (ata27@cam.ac.uk).

**Abstract.** Here we present results from an evaluation of model simulations from the International Hemispheric Transport of Air Pollution Phase II (HTAPII) and Chemistry Climate Model Initiative (CCMI) model inter-comparison projects against a comprehensive series of ground based, aircraft

and satellite observations of ozone mixing ratios made at various locations across India. The study focuses on the recent past (observations from 2008-2013, models from 2008-2010) as this is most pertinent to understanding the health impacts of ozone. To our understanding this is the most comprehensive evaluation of these models' simulations of ozone across the Indian sub-continent to date. This study highlights some significant successes and challenges that the models face in

representing the oxidative chemistry of the region.

      The multi-model range in area weighted surface ozone over the Indian subcontinent is 37.26 – 56.11 ppb, whilst the population weighted range is 41.38 – 57.5 ppb. When compared against surface observations from the Modelling Atmospheric Pollution and Networking (MAPAN) network of eight semi-urban monitoring sites spread across India, we find that the models tend to simulate

higher ozone than that which is observed. However, observations of NOx and CO tend to be much higher than modelled mixing- ratios, suggesting that the underlying emissions used in the models do not characterise these regions accurately and/or that the resolution of the models is not adequate to simulate the photo-chemical environment about these surface observations. Empirical Orthogonal Function (EOF) analysis is used in order to identify the extent to which the models agree with

regards to the spatio-temporal distribution of the tropospheric ozone column, derived using OMI-MLS observations. We show that whilst the models agree with the spatial pattern of the first EOF of observed tropospheric ozone column, most of the models simulate a peak in the first EOF seasonal



cycle represented by principle component 1, which is later than the observed peak . This suggest a widespread systematic bias in the timing of emissions or some other unknown seasonal process.

In addition to evaluating modelled ozone mixing ratios, we explore modelled emissions of NOx, CO, VOCs, and the ozone response to the emissions. We find a high degree of variation in emissions from non-anthropogenic sources (e.g. lightning NOx and biomass burning CO) between models. Total emissions of NOx and CO over India vary more between different models in the same MIP than the same model used in different MIPs, making it impossible to diagnose whether differences in modelled ozone are due to emissions or model processes. We therefore recommend targeted experiments to pinpoint the exact causes of discrepancies between modelled and observed ozone and ozone precursors for this region. To this end, a higher density of long term monitoring sites measuring not only ozone but also ozone precursors including speciated VOCs, located in more rural regions of the Indian sub-continent, would enable improvements in assessing the biases in models run at the resolution found in HTAPII and CCMI.

## 1 Introduction

The issues of increasing levels of surface ozone ($O_3$) and its impacts on human health, the biosphere and climate are of major concern globally. Recent reports (Health Effects Institute, 2017) highlight that ambient ozone contributes to the global health burden through its impact on premature deaths and disabilities from chronic obstructive pulmonary disease (COPD). Nearly 4.5 million people die prematurely each year due to exposure to outdoor pollution, 254,000 of which are due to ozone exposure and its impact on chronic lung disease, the remaining majority are attributed to particulate matter below 2.5µm in diameter ($PM_{2.5}$). Around half of these premature deaths are in China and India (Cohen et al., 2017). However, a recent study using updated risk estimates suggests that previous analyses have underestimated the long-term health impacts of tropospheric ozone, and the true global disease burden could be over one million premature deaths per year, 400,000 of which occur in India (Malley et al., 2017). India and its neighbouring countries, China, Pakistan and Bangladesh, have experienced the largest increase in seasonal average population-weighted ozone concentrations over the last 25 years (Health Effects Institute, 2017), with India alone accounting for 67% of the global increase in ambient ozone attributable deaths due to COPD between 1990 and 2015.

The ill effects of ozone are not only limited to human health. Ghude et al., (2008) calculated relative agricultural yield loss using accumulated ozone exposure exceedances over a threshold of 40 ppb from the analysis of seven years of data of hourly surface ozone concentrations over India (1997-2004) during the pre-monsoon season. They estimated yield losses of 22.7%, 22.5%, 16.3% and 5.5% for wheat, cotton, soya bean and rice respectively, sufficient to feed about 94 million people and an economic value of more than a billion USD per year.





Identifying the sources and sinks of tropospheric ozone and its precursors, and in turn identifying the ways to reduce ambient ozone exposure, remains a key challenge. Ozone is a secondary pollutant, meaning it is not directly emitted into the atmosphere. The tropospheric chemistry of ozone and its precursor species, such as Volatile Organic Compounds (VOCs), carbon monoxide (CO) and nitrogen oxides (NOx = NO + $NO_2$), is complex and involves a large number of species that participate in a cascade of NOx-catalysed chemical reactions that ultimately oxidise VOCs to $H_2O$ and $CO_2$, generating ozone as a by-product (Jenkin and Clemitshaw, 2002; Monks et al., 2015). India is experiencing a rapid growth in its industrial and economic sectors with increasing emissions of pollutants and trace gases associated with this development (Ghude et al., 2008, 2013). An increasing trend in tropospheric ozone over most parts of India has been observed in long-term decadal trend analysis (1979-2000) using satellite based approaches to determine the Tropospheric Ozone Residual (TOR), with the strongest trends observed over the Indo-Gangetic Plain (The IGP region - a region to the north of India, at the foothills of Himalayas) (Lal et al., 2012).

Meteorological parameters also play an important role in driving tropospheric ozone chemistry, as has been demonstrated in many studies in the last few years. Central to the production of ozone is photolysis (photo-dissociation). The presence of clouds can greatly impact the rates of photolytic reactions and so act as a limit for ozone production (Voulgarakis et al., 2009). Ozone also tends to have a positive correlation with temperature and a negative correlation with relative humidity (Camalier et al., 2007). Increases in water vapour directly lead to ozone loss through the reaction of excited oxygen atoms, formed from ozone photolysis, with water, and indirectly through the wet scavenging of compounds which act as reservoirs and precursors for ozone (Monks et al., 2015). These meteorological factors are of particular importance for the Indian sub-continent, where the seasonal cycle is dominated by the monsoon season, lasting for 4 months from June to September and characterised by high precipitation rates, cloudy days, seasonal reversal of prevailing wind directions, and mixing of the clean marine boundary layer air from south-west with the continental air. Ground based studies on ozone cycles at various sites in India report that the minimum ozone values observed during the monsoon season are likely attributed to high relative humidity, low solar radiation, cloudiness conditions and wet scavenging of ozone precursors. In contrast, the high temperatures, high solar radiation and low humidity during the pre-/post-monsoon seasons provide favourable conditions for photochemical production of $O_3$. During winter, low temperatures, low solar radiation and fog limits the photochemical $O_3$ production in most parts of India. (Beig et al., 2007; Sinha et al., 2015; Yadav et al., 2016). An exception is the Mt. Abu site in northern India. Due to the unique meteorology at this high altitude site, the seasonal variation in surface ozone shows a maximum in late autumn and winter (Naja et al., 2003).

Owing to the complex interplay between emissions, chemistry and the unique meteorology that impacts the Indian sub-continent, and the limited coverage of surface observations, three-dimensional numerical models are required to estimate the health burden of ozone exposure and



predict how ozone levels will respond to future changes in emissions and climate. Three-dimensional numerical models include meteorology, emissions and complex photo-chemical mechanisms to simulate ozone concentrations (Keeble et al., 2017; Surendran et al., 2015). But these models need to be evaluated with as many observations of as many species that contribute to

ozone production and loss as possible. The ability of a model to accurately predict the present state of species gives us the confidence to rely on them for future projections as well as to predict the levels of pollutants in regions where observations are limited. Many previous studies have evaluated the ability of chemistry-transport models to simulate levels of ozone and other key species for tropospheric chemistry over North America and Europe, where dense, long term and reliable

measurements are available (Im et al., 2015; O'Connor et al., 2014; Tilmes et al., 2015). Owing to the sparsity of in situ data, these kinds of studies are limited over the Indian-subcontinent. Evaluation of models and their agreement as well as disagreement over this region will enhance our understanding about the production of ozone and the factors controlling it. An improvement in our fundamental ability to simulate the processes which control ozone will ultimately enable the best

policy decisions to mitigate the impacts of ozone on human health and crops in the region.

In this paper, we have evaluated model simulations from the international Hemispheric Transport of Air Pollution Phase-II (HTAPII) and Chemistry Climate Model Initiative (CCMI) model inter-comparison projects against a comprehensive series of ground based, aircraft and satellite observations of ozone, NOx and CO across India. To our knowledge, this represents the most

exhaustive evaluation of ozone for these models in this region and enables us to characterise seasonal biases and errors between the models. Section 2 describes the models that we have used in these analyses and the observations we used to evaluate the models against. In section 3 we present the results of our evaluation, including Empirical Orthogonal Function (EOF) analysis to identify similarities and differences in the spatio-temporal distribution of the tropospheric ozone

column simulated in the models and retrieved from the OMI-MLS instruments (Ziemke et al., 2011). In Section 4, we discuss the results and suggest possible future research needed to understand ozone chemistry over the Indian subcontinent.

## 2 Methodology

### 2.1 Datasets for evaluation

#### 2.1.1 Ground based Observations

The model simulations have been validated against measurements of surface ozone from eight stations located across India in: Delhi, Patiala, Udaipur, Jabalpur, Pune, Hyderabad, Guwahati and Chennai. Figure 4 shows the geographical locations of these stations. Details of all the ground

based stations have been summarised in Table 1. The coordinated measurements of trace gases and aerosols at these locations of India are carried out under the Indian Institute of Tropical Meteorology (IITM), Pune, India and Ministry of Earth Sciences (MoES) as part of the 'Modelling





Atmospheric Pollution and Networking' (MAPAN) programme. All the monitoring stations are designated as semi-urban indicating that the stations are away from downtown areas where the influence of local emissions may be very high. However, as we show in section 3, these are far from pristine measurement locations and appear to be influenced by high levels of NOx and CO. Observations at these stations were made with the Air Quality Management System (AQMS). The AQMS comprises of US Environmental Protection Agency approved analysers housed inside walkway shelters and have a sampling height of 3 meters above ground level (Beig et al., 2013).

**Table 1**: Details of the locations of in situ ozone monitoring stations used in this study. All stations are categorised as semi-urban sites. All data were collected at an hourly resolution throughout the year 2013. For more details see section 2.1.1.

| Stations | Latitude (°N) | Longitude (°E) | Elevation (meters above sea level) | Institutes |
|---|---|---|---|---|
| Delhi | 28°41' | 77°12' | 253 | IMD, Lodhi road |
| Patiala | 30°21' | 76°22' | 257 | Thapar University |
| Udaipur | 24°35' | 73°43' | 255 | M.L.S University |
| Jabalpur | 23°9' | 79°58' | 420 | Govt. Model Science College |
| Pune | 18°32' | 73°48' | 590 | IITM, Pune |
| Hyderabad | 17°31' | 78°24' | 609 | INCOIS |
| Guwahati | 26°9' | 91°39' | 56 | Gauhati University |
| Chennai | 13°2' | 80°8' | 20 | Sri Ramchandra University |

The measurements of surface $O_3$, NOx and CO were made continuously at hourly time resolution during the year 2013. Ozone measurements were conducted using an Ecotech Ozone analyzer (model number EC 9810B), which combines the benefits of microprocessor control with ultraviolet (UV) photometry at 254 nm to accurately measure ozone mixing ratios in ambient air. The analyzer provides accurate measurements of ozone in the range of 0–20 ppmv with a detection limit of 0.5 ppbv and has a linearity error of less than 3%.



The measurements of NOx were performed by using an Ecotech Nitrogen Oxides Analyzer (model number EC 9841B). This analyzer works on the chemiluminescence technique for accurate and reliable measurements of NO, $NO_2$ and NOx mixing ratios. The technical limitations (artifacts) of the chemiluminescent based methods have been well reported (Fuchs et al., 2009; Winer et al.,

1974). CO was measured using Ecotech, model EC 9830 analyzer based on the infrared (IR) photometry. Information about the maintenance and calibration of these instruments have been reported earlier (Chakraborty et al., 2015; Yadav et al., 2014). Monthly mean values for $O_3$, NOx, CO were calculated from the 24-hour averages of the hourly data. Days with fewer than 15 hours of observations were excluded from the analysis.

**2.1.2. CARIBIC Observations**

The CARIBIC project (Civil Aircraft for the Regular Investigation of the atmosphere Based on an Instrument Container, www.caribic-atmospheric.com) aims to investigate the spatial and temporal distribution of a wide-range of compounds. It is based on the use of a fully automated scientific

instrument package in a 1.5t container aboard a passenger aircraft which is equipped with an advanced multi-probe inlet system (Brenninkmeijer et al., 2007). In the region of interest, flights operated monthly from April to December 2008 aboard a Lufthansa Airbus A340-600 passenger aircraft flying from Frankfurt to Chennai. The total number of flights during this period was 16. Usually one set of flights consisted of four consecutive flights, i.e. two round trips from Frankfurt to

Chennai within three days, with exception of July and October, when only one round trip was performed. The ascents and descents of the flights took place during night, with landing times around 23:30 local time and take off times around 02:00 and 03:40 local time the next morning (Ojha et al., 2016).

The ozone measurements were made by a dry chemiluminescence (CL) instrument, which at

typical ozone mixing ratios between 10 ppb and 100 ppb and a measurement frequency of 10 Hz has a precision of 0.3–1.0 %. The absolute ozone concentration is inferred from a UV-photometer designed in-house which operates at 0.25 Hz and reaches an accuracy of 0.5 ppb. The CL instrument has been discussed in detail by Zahn et al., 2012.

CO is measured with an AeroLaser AL 5002 resonance fluorescence UV instrument modified

for use on board the CARIBIC passenger aircraft. The instrument has a precision of 1– 2 ppbv at an integration time of 1 s and performs an in-flight calibration every 25 min. Technical details of the CO instrument can be found in Scharffe et al., 2012.

The CARIBIC observations taken during ascent as well as descent of the flight have been considered in this study. These observations are averaged into vertical bins of 25 hPa. For monthly

mean vertical profiles, average of all the ascending and descending profiles during that month have been considered. For comparison, monthly mean model simulated profiles over Chennai are also averaged into vertical bins of 25hPa and have been interpolated to the CARIBIC pressure levels.





### 2.1.3. OMI/MLS Tropospheric Column Ozone (TCO) measurements

Tropospheric Column Ozone (TCO) for the year 2010 is derived using the tropospheric ozone residual (TOR) method, which is the residual of total column ozone from Ozone measuring instrument (OMI) and stratospheric column ozone from Microwave Limb Sounder (MLS) with spatial resolution of Aura/MLS (Ziemke et al., 2011), Schoebert et al. 2007). TOR is an integrative product which accounts for changes in ozone not only at the surface, where it is most detrimental to human and crop health, but also the free troposphere, where it has a longer lifetime and so is influenced by more sources and has a larger climate impact (Stevenson et al., 2013).

OMI and MLS are two out of four instruments on board the Aura satellite, which orbits the Earth in sun synchronous polar orbit at 705 km altitude and 98.2° inclination. OMI is a nadir viewing instrument which detects back scattered solar radiance from Earth at visible (350-500 nm) and UV (270-314nm, 306-380nm) wavelengths to measure total column ozone with a spatial resolution of 13km X 24km. The MLS instrument detects microwave thermal emissions from the limb of Earth's atmosphere to measure mesospheric, stratospheric and upper tropospheric temperature, ozone and other constituents. MLS measurements are taken about 7 minutes before OMI views the same location during ascending (daytime) orbital tracks. Details of these instruments are discussed elsewhere (Waters et al., 2006).

### 2.2 Model Description

In this work we aim to evaluate how a range of models perform over the Indian sub-continent to understand what the level of agreement in ozone modelling is, in this traditionally observation poor area. We focus here on global models as these are increasingly used in assessments of the health impacts of air pollution (e.g. Malley et al., 2017; Lelieveld et al., 2018). There is a long history of co-ordinated Model Intercomparison Projects (MIPs), with the general aim of co-ordinating modelling centres to better understand how the state-of-science models compare against each other and observations. MIPs are generally focused on specific science questions which define the length of the integrations performed with the models and the amount of model output requested. MIPs have been the key mechanism to bring together our understanding of climate change and are increasingly enabling our understanding of atmospheric composition to be improved.

**Table 2**: Description of the eight global chemistry climate models used in this study. The Table also gives Global emissions of NOx, CO and Global Tropospheric Ozone burden simulated by each model.





| Model Name | Abbr. | MIP | Institution | Version | Experiment | Resolution lat x lon x No. Of vertical levels | References | Global Emissions | | Global Tropospheric ozone burden Tg/year |
|---|---|---|---|---|---|---|---|---|---|---|
| | | | | | | | | NOx Tg(N) /year | CO Tg(CO) /year | |
| HadGEM2-ES | HDGM | HTAPII | Met Office and Univ. of Cambridge, UK | | BASE_2009 | ~1.25° x 1.875° x 38 | (Collins et al., 2011) | 37.5 | 978.5 | 379.6 |
| GEOSCHEM–ADJOINT | GCAD | HTAPII | Univ. of Colorado, Boulder | | BASE_2010 | ~2° x 2.5° x 47 | (Henze et al., 2007) | 54.3 | 1001.3 | 340.7 |
| CHASER | CHSR | HTAPII | Nagoya Univ., JAMSTEC | CHASER-V4 MIROC–ESM | BASE_2010 | ~2.76° x 2.8° x 32 | (Sudo et al., 2002) | 49.7 | 915.05 | 318.5 |
| MOZART4 | MOZT | HTAPII | Indian Institute of Tropical Meteorology, India | MOZART4 | BASE_2010 | ~1.89° x 2.5° x 56 | (Surendran et al., 2015) | 44.2 | 1014.1 | 358.1 |
| MRI-ESM1r1 | MRIE | CCMI | Meterological Research Institute, Japan | *r1i1p1,v1 | REFC1_2010 | ~2.7° x 2.8° x 80 | (Adachi et al., 2013) | 55.47 | 1172.4 | 384.8 |
| GEOSCCM | GCCM | CCMI | NASA Goddard Spaceflight Centre, USA | *r1i1p1,v3 | REFC1_2010 | ~2° x 2.5° x 72 | (Oman et al., 2011; Rienecker et al., 2008) | 40.84 | 1176.2 | 336.9 |
| CHASER–MIROC–ESM | CHSM | CCMI | Nagoya Univ., JAMSTEC | *r1i1p1,v1 | REFC1SD_2010 | ~2.76° x 2.8° x 57 | (Sudo et al., 2002) | 43.3 | 908.64 | 326.8 |
| UMUKCA–UCAM | UKCA | CCMI | Univ. of Cambridge, UK | *r1i1p1,v1 | REFC1_2010 | ~2.5° x 3.75° x 60 | (Bednarz et al., 2016; Morgenstern et al., 2017) | 32.76 | 867.31 | 353.1 |

*r=realization number of simulation, i=initialization method, p=perturbed physics, v=version of publication level.

The most recent global MIPs include both the Chemistry Climate Model Initiative (CCMI) (Morgenstern et al., 2017) and International Hemispheric Transport of Air Pollution Phase II

5   (HTAPII). (Koffi et al., 2016) We opted to look at data from both of these MIPs but, owing to constraints on time and data availability, chose to focus on a sub set of models. Specifically, we examine output from simulations from the following eight models:



- HadGEM2-ES model (Collins et al., 2011; Jones et al., 2011), hereafter referred to as HTAPII-HDGM;
- GEOS-Chem Adjoint (Henze et al., 2007), hereafter referred to as HTAPII-GCAD;
- CHASER-v4-MIROC-ESM and CHASER-MIROC-ESM (two different configurations of essentially the same model run for HTAPII and CCMI and referred to as HTAPII-CHSR and CCMI-CHSM respectively) (Sudo et al., 2002a; 2002b);
- MOZART-4 (Divya et al., 2015), hereafter referred to as HTAPII-MOZT;
- MRI-ESM1r1 (Yukimoto et al., 2011; Deushi & Shibata 2011), hereafter referred to as CCMI-MRIE;
- GEOSCCM (Oman et al., 2011; Reinecker et al., 2008; Duncan et al., 2007; Strahan et al., 2007), hereafter referred to as CCMI-GCCM;
- UMUKCA-UCAM (Bednarz et al., 2016), hereafter referred to as CCMI-UKCA.

Table 2 outlines the details of the above models, which MIPs the models were run as part of, and documents our calculations of the tropospheric ozone burden in each model (using a consistent treatment of a chemical tropopause defined using a 150 ppb monthly mean ozone iso-surface). These models span a range of horizontal resolution (lowest resolution is CCMI-UKCA at 2.5° lat x 3.75° lon and highest resolution is HTAPII-HDGM at 1.25° lat x 1.85° lon), vertical resolution (HTAPII-CHSR/CCMI-CHSM have 32 vertical model levels, whilst CCMI-MRIE has 80 vertical model levels), and use chemical mechanisms of differing complexity and scope (e.g. CCMI-UKCA has been designed for simulations of mainly stratospheric nature whilst HTAPII-CHSR/CCMI-CHSM use a chemistry scheme much more focused on tropospheric oxidation with a larger number of non-methane VOCs). The lowest model level varies from a minimum of 25 m for CCMI-MRIE to 124 m for HTAPII-GCAD. For further details of the model set ups please see the cited references for each model in Table 2 and the MIP description papers (i.e. for the CCMI models see Morgenstern et al., 2017). From our analysis of the tropospheric ozone burden, we see that all models lie within the range of the Atmospheric Chemistry and Climate Model Intercomparison Project (ACCMIP) models (Young et al., 2013) and the likely range as recently quantified through satellite retrievals of the tropospheric column analysed by the IGAC Tropospheric Ozone Assessment Report (TOAR) (Gaudel et al., 2018).

From the eight models described above, we focus our analysis on monthly and daily mean mixing ratios of ozone, NOx and CO, and monthly mean surface emissions of CO, NOx and lightning derived NOx. We focus on output from the models appropriate for the year 2010 and limit the main analysis to the domain of 56° to 105° longitude and 5° to 38° latitude, which covers the entire Indian Subcontinent.

In spite of simulating the same period of time, CCMI and HTAPII use different base emission inventories as part of their protocol. Surface CO and NOx emissions, which over the Indian sub-continent are dominated by anthropogenic sources, should generally be consistent within MIPs,



which we largely see but explore in more detail below. Lightning is an important source of NOx to the remote atmosphere, and an emission term that tends to not be possible to specify in MIPs, so reflects an area of emissions that models should, and do, differ in and an aspect we assess in more detail below.

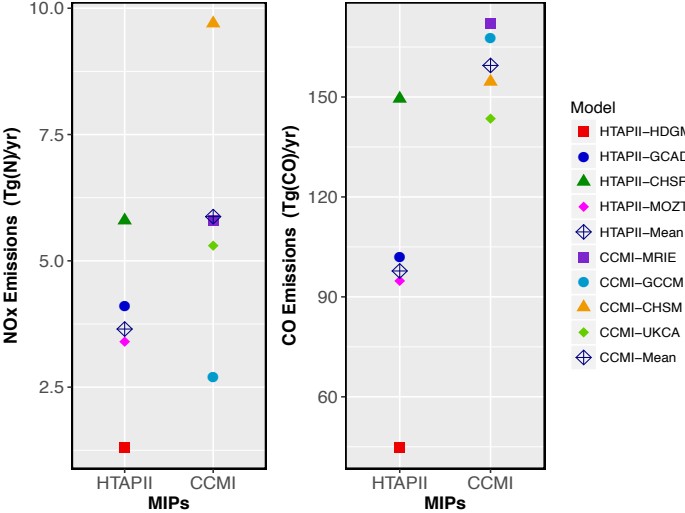

Figure 1: Shows the high variability in NOx and CO emissions between the two MIPs over the domain considered in this study. CCMI models show larger variability for NOx emissions and HTAPII models shows larger variability for CO emissions.

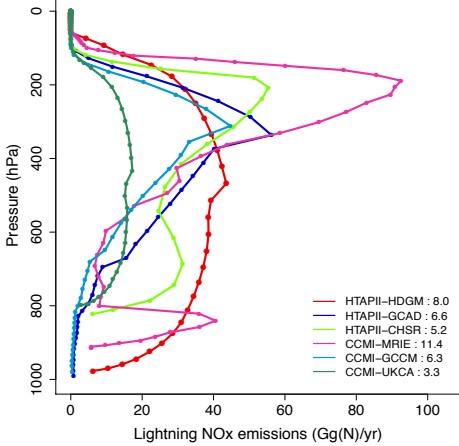

Figure 2: Shows the annual vertical profiles of lightning NOx emissions over the domain considered in this study. (inset) Global emissions of Lightning NOx in Tg(N)/yr as simulated by each model.



### 2.2.1 Description of emissions from model simulations

The annual total NOx and CO emissions for all models over the domain are shown in Figure 1 and in supplementary figures S2 and S3. Briefly, there is large variability in input emissions of NOx and CO for the different models and MIPs (S2 and S3). The intra MIP variability is greater than the inter

MIP variability for NOx i.e. more variability within a MIP for NOx emissions than between them (see Figure 1). However, the converse is true for CO where the CCMI emissions tend to be higher than those used in the HTAPII MIP. For individual MIPs, every modelling group was required to use the same anthropogenic emissions data. Disparities in emissions may be due to the use of different natural and biomass burning sources.

Lightning is the largest contributor to upper tropospheric NOx and it is a source of largest uncertainty. Global emissions of lightning NOx (LNOx) as simulated by the models show a variance of 7.56 Tg(N)/yr (annual global emission of LNOx as simulated by each model is given in Figure 2). The vertical profiles of LNOx emissions are very different in each models over the domain considered in this study (Figure 2). Parameterisation of LNOx is highly dependent on the vertical

and horizontal resolution of the models. CCMI-UKCA and HTAPII-HDGM models show similar vertical profiles as they have a similar internal configuration. The difference in the convection parameterisation in these models lead to a difference in the magnitudes of LNOx emissions. CCMI-MRIE clearly stands out giving highest values of LNOx emissions globally as well as over the Indian subcontinent.

### 3 Results

Here we evaluate four model simulations each from HTAPII and CCMI, with a set of ground based, satellite and airborne observations of $O_3$, ground based and airborne observations of CO and ground-based observations of NOx.

### 3.1 Annual mean model simulated surface ozone



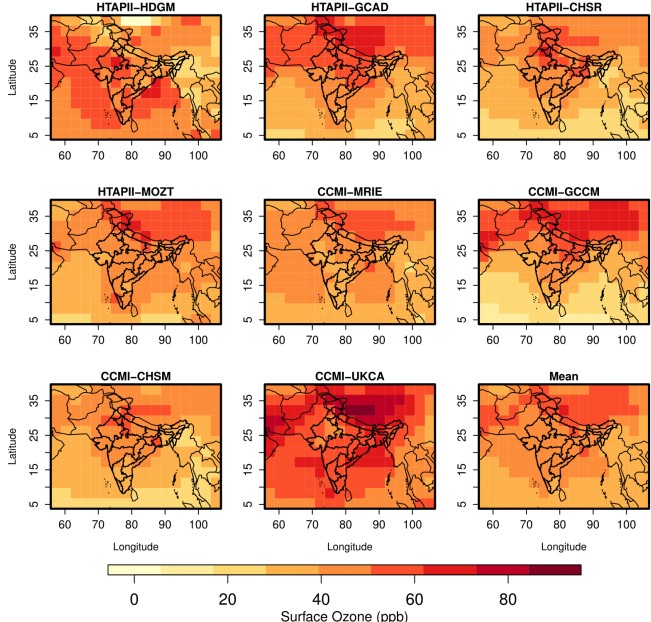

Figure 3: The spatial patterns of annual mean surface ozone (ppb) as simulated by the lowest level in each model highlighting the regions where the models show maxima and minima over the Indian Subcontinent.

Figure 3 shows the spatial patterns of annual mean surface ozone mixing ratios from the model simulations described in Table 2 and the multi model mean (MMM), shown in the lower right-hand panel. Ozone mixing ratios from the lowest model level are considered as surface ozone in this study. There is general agreement in the spatial characteristic of annual mean surface ozone across the models, except for HTAPII-HDGM (it shows different maxima and minima as compared to the

other models). The range in area weighted surface annual mean ozone is 22.9 - 35.3 ppb, with HTAPII-CHSR at the lower and HTAPII-MOZT at the upper end of the range, and the MMM value is 29.3 ppb. We also investigated the populations weighted surface annual average statistics using population data from NCAR climate and global dynamics (Gao, n.d.; Jones and O'Neill, 2016). These data have a range of 28.5 - 38.85 ppb, with HTAPII-CHSR at lower end and CCMI-UKCA at

the upper end and a MMM of 33.0 ppb.

The MMM shows that the highest values of surface ozone are over the Tibetan plateau and northern part of India and the lowest values over the southern peninsula. However, whilst the models broadly agree on the regions of higher and lower ozone, there is significant intermodal variability in magnitude of ozone concentration. Variations in models can be attributed to the

different chemical schemes, physical parameterisations, grid resolution and non-anthropogenic emissions used in the models. CCMI-UKCA shows the highest values of surface mean annual average ozone compared to the other models. This may be attributed to the fact that CCMI-UKCA



was designed for stratospheric chemistry hence, contains only a limited set of tropospheric chemistry reactions and no isoprene chemistry (more details in section 2.2).

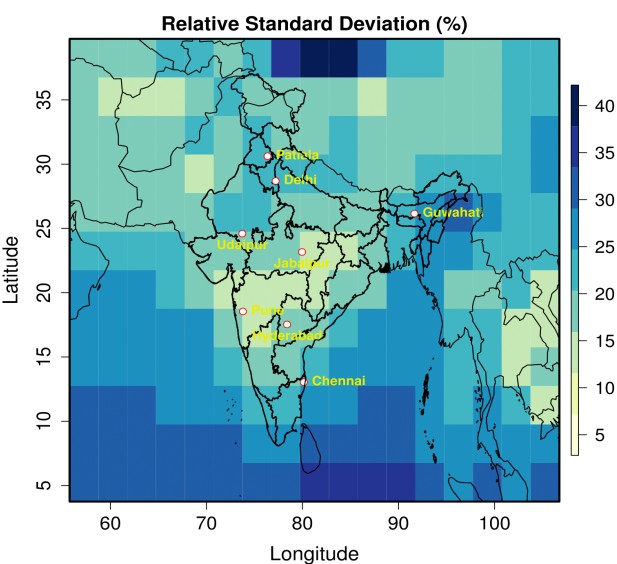

Figure 4: Relative standard deviation of surface ozone from the eight models. The plot also shows the location of ground based observational MAPAN-stations considered in this study.

The standard deviation of the multi model ensemble is shown in Figure 4. The standard deviation of the multi model mean can be used as an indicator of the level of agreement between the models. Here we show that there is a reasonably low level of agreement between the models, with an average of 23% standard deviation in the mean. This is worse than the level of agreement between the ACCMIP models (over northern and southern hemispheres with most observations in Europe and North America) shown in Young et al., 2013 and could reflect the fact that here we compare simulations from two different MIPs which make use of different emissions. However, we find the difference between the emissions within models of a particular MIP is as large as those between MIPs (Figures 1, S2 and S3).  Figure 4 highlights that models differ most in the northern and eastern part of India and standard deviation is the least in the central part of India.

**3.2 Comparison between Models and ground based surface observations**

**3.2.1 Ozone**

Comparison of model simulated monthly mean surface ozone with the monthly mean of hourly observations from the eight ground based monitoring stations listed in Table 2 is shown in Figure 5. In contrast to locations in Europe and North America, but in agreement with previous observational analysis of surface ozone over India (Beig et al., 2007; Jain et al., 2005; Lal et al., 2012), our observational data highlight a double peak structure in the seasonal cycle of surface ozone. Cloudiness and wet scavenging of ozone precursors during the monsoon period (June-September)





limit the photochemical production of ozone, resulting in lower values of ozone during these months. Due to favourable meteorological conditions during pre- (April-May) and post- (October-November) monsoon seasons, such as strong solar radiation, high temperature and low humidity, photochemical production of ozone is enhanced during these months. Emissions from biomass burning also contribute to ozone production during the post-monsoon season at sites such as Delhi and Patiala. The seasonal variability in the models is captured fairly well at all stations, except at Chennai. Figure 5 includes the MMM and standard deviation (dark dashed blue and light blue envelope), which can be compared with the mean and standard deviation of the observations (solid black line and grey envelope). In all locations, the ozone mixing ratio is higher in the MMM than in the observations. The overestimation by the models is due to the overestimation in production and/or the underestimation of loss of ozone. This could be attributed to a combination of factors. The principal factor is most likely a miss match in the representativeness of the observational sites for comparison with the coarse resolution models. At coarse resolution, the models cannot capture fine-scale processes, such as the impact of nearby sources of pollution (e.g. NOx emissions) on the observations of ozone. Ozone production is highly non-linear in terms of the precursor emissions VOCs and NOx (Monks et al., 2015). Figure 5 also highlights differences between the models. There is considerable inter-model variation in simulating the seasonal variation in surface ozone, as we discuss in more detail below.

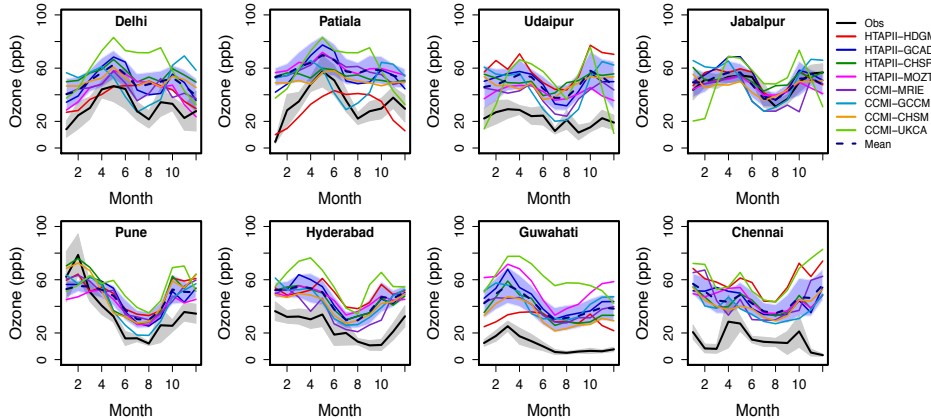

Figure 5: Comparison between ground based observations, model simulated and the ensemble mean of monthly mean surface ozone over the eight MAPAN-Stations.

To evaluate the performance of models at each station we compare the Normalised Mean Biases (NMB) and Pearson Correlation Coefficient (R). These were calculated using following equations:

$$\text{NMB} = \frac{\sum \text{Model} - \sum \text{Obs}}{\sum \text{Obs}} \qquad\qquad (1)$$





$$R \quad = \quad \frac{\overline{(\text{Model}[\iota] - \overline{\text{Model}}) * (\text{Obs}[\iota] - \overline{(\text{Obs})})}}{\sigma(\text{Model}) * \sigma(\text{Obs})} \tag{2}$$

where, σ is the standard deviation.

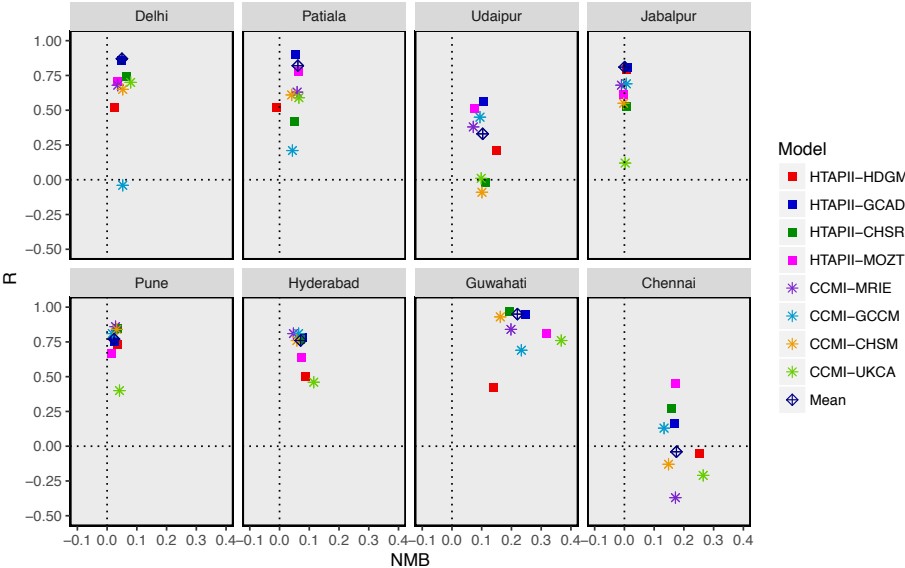

Figure 6: Scatter plot of Pearson Correlation Coefficient (R) values against Normalised Mean Bias (NMB) highlighting the models performance in surface ozone at each station.

Figure 6 shows the relationship between R and NMB for each of the models we have studied, as
well as the multi model mean, at each of the surface site locations. As is evident from Figure 5, all
models show a positive NMB at all stations. All models have low biases and high R-values (except
CCMI-UKCA) at Jabalpur and Pune. Models show high biases at Guwahati and Chennai, and low R-
values at Chennai and Udaipur. Overall, the performance of the models across all the sites is
inconsistent. There is no one model that performs systematically well at all stations. Conversely, the
models perform differently at each station in terms of their R-value and NMB. Unlike in previous
studies (e.g. Young et al., 2013) the MMM also does not outperform the individual models in Figure
5. CCMI-UKCA acts as an outlier at 5 out of 8 sites. The impact of the underlying emission biases
can be seen by comparing the results between HTAPII-CHSR and CCMI-CHSM in this study. These
are in effect the same model (see section 2.2) but include different emissions data as part of the
different MIP protocols. Figure 6 shows that these two simulations result in large differences at only
one of the 8 sites investigated (Chennai), whereas the difference between different models in the
same MIP is typically much larger. This implies that the differences between the simulations are
more down to the differences in model setup, representation of chemical and physical processes




and non-anthropogenic emission sources in the models than the differences in anthropogenic emissions between the two MIPs.

In order to better understand the causes of biases between the model and observations shown in Figures 5-6, 24-hour average model and observation data have been analysed to
determine probability density functions (PDF) as shown in Figure 7 for a sub set of the sites considered (Delhi, Pune, Guwahati and Chennai). The PDFs for the observations show a multimodal distribution (with 2-3 modes most common) with the highest peak at lower ozone values. This pattern is typical of situations where nearby sources of NOx titrate ozone, through the reaction:

$O_3 + NO \rightarrow NO_2 + O_2$                                    (3)

The observed PDFs are typically low in Guwahati and Chennai, whereas Delhi and Pune show several days where high levels of ozone are seen, especially in Pune where daily average ozone can be as high as 97 ppb.

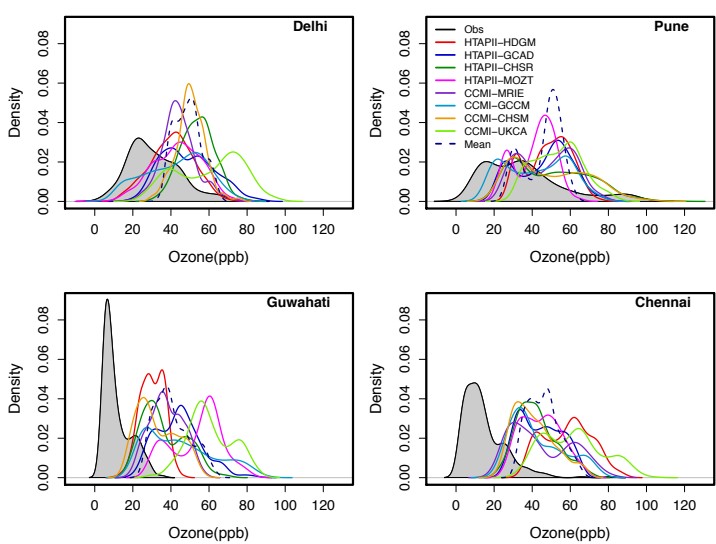

Figure 7: Probability Density Functions (PDF) for in situ observations and model simulated 24 hour average surface ozone at Delhi, Pune, Guwahati and Chennai.

The PDF for the model simulations also show a multimodal distribution but the nature of their distributions is very different from the observed distribution. Moreover, the differences between the simulation PDFs is larger than the differences between the multi model mean and the observations. Again, this highlights the large variability between models in their simulation of ozone in these regions. The most obvious feature from Figure 7 is that the models overestimate the PDFs at the
four sites and significantly overestimate the tails of the ozone distributions. As well as overestimating



the ozone concentration at the modes, in most models the highest peak is at the second mode with higher ozone values, in contrast to the observations where the highest peak is usually at lower ozone concentrations. The amplitude and shift in the PDF peaks compared to observations is greatest at Guwahati and Chennai. This may be due to the inability of the models to adequately

simulate NOx titration at these sites, which occurs at a finer scale than can be resolved by the coarse model grids. Studies have shown that the model's ability to simulate surface ozone is very sensitive to horizontal resolution and high resolution model generally perform better when compared to observations (Stock et al., 2014).

**3.2.2 Ozone precursors**

When compared with the set of available surface ozone observations we have used, the current state-of-the-art global chemistry models overestimate surface ozone in India. There is a large amount of variability between the models, much larger over India than in previous model inter-comparisons over northern and southern hemispheres (Young et al., 2013). In order to better

understand the variation of ozone, we have also compared the model simulations of NOx and carbon monoxide at the eight sites that form part of the MAPAN network. Similar to Figure 5, Figure 8 shows the seasonal variation in surface NOx in the models and observations. The observations of NOx (black line with grey envelope) vary from location to location. High values are observed during autumn/winter due to the transport of pollutants from polluted regions, such as the IGP region,

through northeasterly winds. During the winter months NOx emissions are trapped closer to the surface due to low boundary layer heights, caused by frequent temperature inversions, while in summer months southwesterly winds bring in clean marine air to almost the entire Indian region and there is greater mixing with free tropospheric air, causing dilution of pollutants in general (Beig et al., 2007; Jain et al., 2005; Lal et al., 2012). Figure 8 shows that in Pune, Guwahati and Jabalpur, the

highest observed monthly average NOx is seen in the winter months. In Delhi and Patiala, the pre- and post-monsoon season (when biomass burning is high) are when NOx levels are at their highest levels, with lower levels of NOx in the monsoon months. Figure 8 highlights that there is a large range of NOx values in the observations, with Delhi having the largest monthly average levels of NOx of up to 180 ppb (November) and Chennai having the lowest levels of NOx (8 ppb, November).





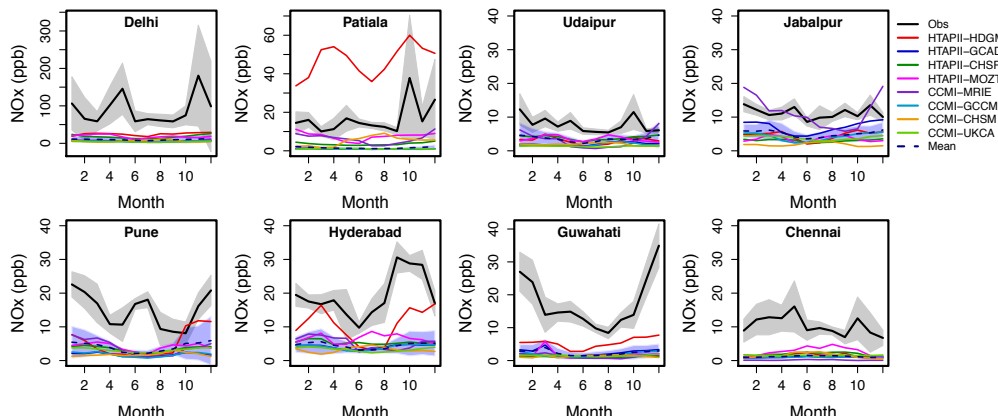

Figure 8: Comparison between ground based observations and model simulations of monthly mean surface NOx over the eight MAPAN-Stations. Different scale has been used for Delhi and Patiala.

5    Comparing the observations and the MMM highlights that on an average the simulations underestimate levels of NOx at these 8 locations across India. An exception is for HTAPII-HDGM at Patiala, where the model simulation overestimates the levels of NOx present. The monthly average NOx in the model simulations at all sites is dominated by $NO_2$ whereas in observations at Delhi (the only site for which separate measurements of NO and $NO_2$ are available), NO dominates monthly

10   average NOx (see section S4 of supplementary material). This discrepancy could be attributed to the coarse resolution of the models meaning high NOx emissions are diluted over larger volume of air. Hence, models underestimate ozone titration due to high levels of NO near emission sources, which results in the overestimation of surface ozone and a photostationary state with greater proportion of NOx as $NO_2$.

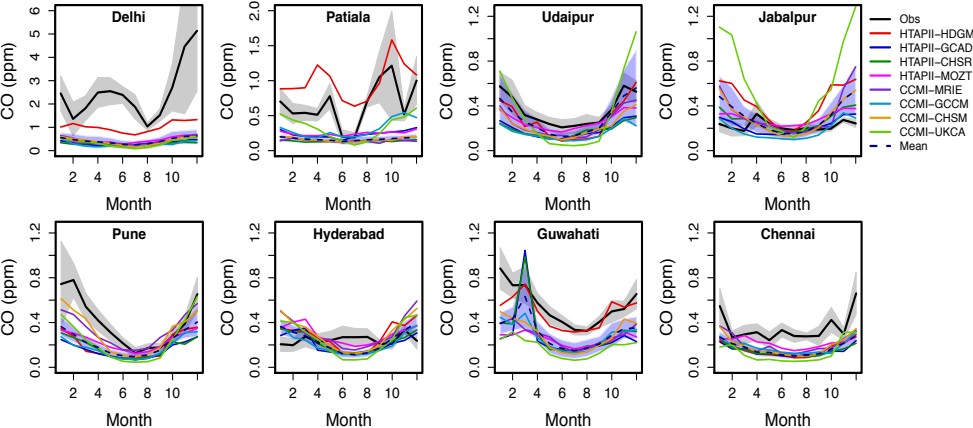

Figure 9: Comparison between ground based observations and model simulations of monthly mean surface CO over the eight MAPAN-Stations. Different scale has been used for Delhi.



Figure 9 shows a comparison of the observed and model simulated CO at the different sites across India we focus on here. At the majority of the other sites considered, there is a clear seasonal cycle in CO with peaks in the winter months and minimum values during the summer and monsoon period. As with NOx (Figure 8), Delhi is the region with highest observed values of carbon monoxide and the highest levels of CO occur in the pre- and post-monsoon period (consistent with the periods of highest agricultural burning). The variations in observed carbon monoxide are caused by a combination of factors including changes in the strength of direct emissions of CO, as well as the contribution of secondary sources such as oxidation of VOCs (Grant et al., 2010), variations in the boundary layer height and changes in local wind patterns (Ahammed et al., 2006).

The model simulations capture the seasonal variability in monthly mean CO well (R-values > 0.4 for all models) at most locations; the exception is in Hyderabad where all models generally show a negative correlation with the observations and at Jabalpur where correlation is poor (see section S5 of supplementary material). The site with the best correlation is Udaipur, where the MMM correlation coefficient is 0.96. Models show good correlation with the observed CO at all sites but highly underestimate the observed values at Delhi and Patiala. As with NOx, an exception is HTAPII-HDGM, which tends to overestimate CO at Patiala, but with good correlation (R-value of 0.63), picking up the peaks in CO pre- and post-monsoon associated with burning.

**3.3 Comparison between models and satellite data**
**3.3.1 Annual Average Tropospheric Ozone Column (AATOC)**





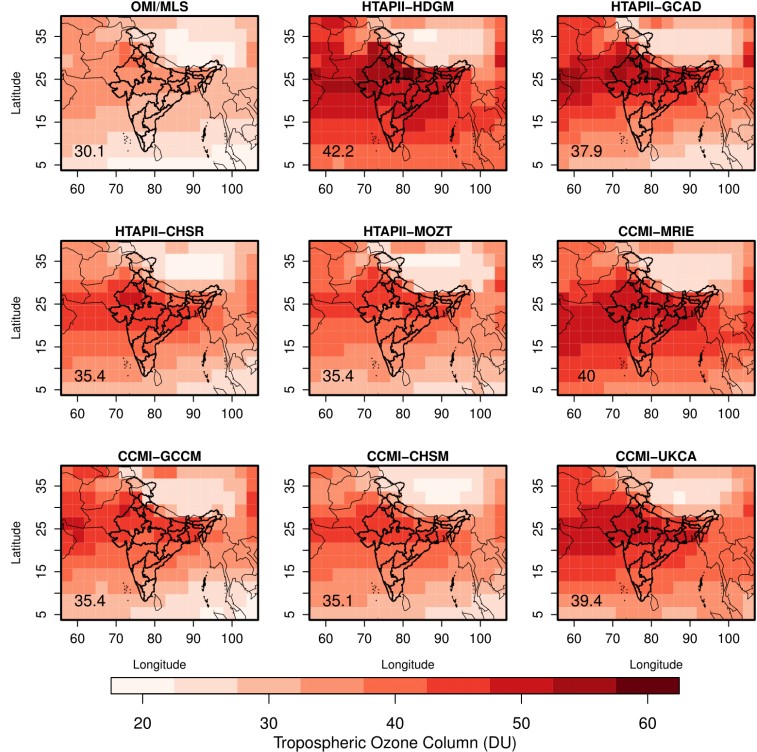

Figure 10: OMI/MLS determined and model simulated Annual Average tropospheric ozone column (AATOC) in Dobson Unit (DU) over the Indian sub continent. Values in the Left bottom corner indicate the mean AATOC in the domain.

Figure 10 shows the annual average tropospheric ozone column (AATOC) retrieved by the OMI/MLS for the year 2010 on board the AURA satellite and the model simulations. The OMI/MLS AATOC shows highest values (45-60 Dobson Units) over the IGP, central and northwestern regions of India. These high values are not uncommon globally (Gaudel et al., 2018). High levels of AATOC are associated with high anthropogenic activities and large scale biomass burning. The IGP and the

regions of India mentioned above are examples of regions affected by these sources. Lower values of AATOC are observed over the maritime regions and a minima is observed over the Tibetan plateau. The seasonal cycle of TOC peaks in May-June and is fairly widespread over India. The onset of the monsoon leads to lower levels of TOC across the region on the whole. Hence, differences in emissions are not the only factor that leads to differences in the observed AATOC

values; regional variations in meteorological conditions are also an important factor that controls AATOC (David and Nair, 2013).

In order to evaluate the model simulations and observations we first compare the mean tropospheric ozone column (MTOC). Over the entire region we focus on (56° to 105° longitude and 5° to 38° latitude), the MTOC from OMI/MLS is 30.1 DU. Models overestimate the MTOC over this

region (see Figure 10.) with MTOC values for models ranging from 35 - 42 DU. HTAPII-HDGM



shows the highest bias (~40%) and HTAPII-CHSR, HTAPII-MOZT and CCMI-GCCM shows lowest bias (~16%). It is worth noting that the AATOC values are not the highest for CCMI-UKCA, even though the annual average surface ozone values are the highest for CCMI-UKCA as compared to the other models.

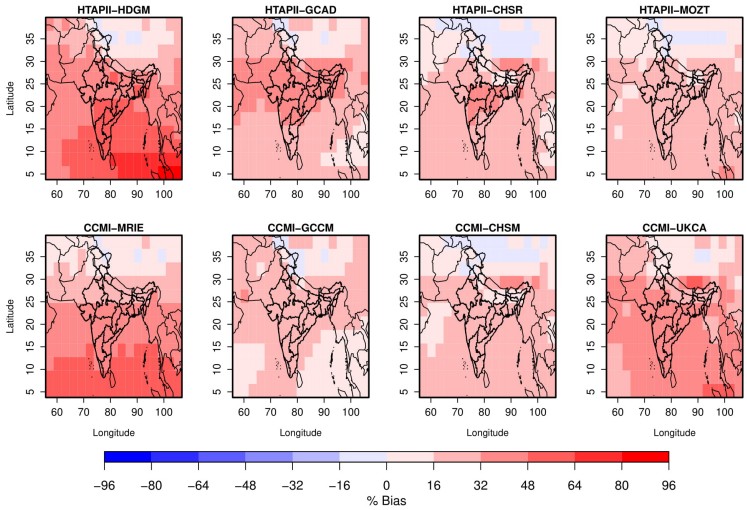

Figure 11: Percentage Biases in model simulated AATOC with respect to the OMI determined AATOC.

The differences between the OMI/MLS observations and the model simulations are further highlighted in Figure 11 where the percentage biases in AATOC are shown. The model simulations, in general, show similar spatial patterns in AATOC to OMI/MLS, but all models overestimate the total TOC values over the domain. The total TOC values for HTAPII-CHSR and CCMI-CHSM are somewhat different and show different bias patterns in spite of having same chemistry schemes and being based on the same model. However, the differences between different models in the same

MIP (i.e. between HTAPII-CHSR and the other HTAPII models, or between CCMI-CHSM and the other CCMI models) are typically larger, both in terms of the average total TOC over the domain and the spatial distribution. Thus, there is greater inter-model variation due to model setup (either differences in model chemistry schemes, dynamics or non-anthropogenic emission sources) than due to differences in anthropogenic emissions prescribed by the two MIPs.

### 3.3.2 Empirical Orthogonal Function Analysis

Several previous studies have focused on harmonic or spectral analysis of time-series' of ozone in both observations and models (Bowdalo et al., 2016; Derwent et al., 2013; Parrish et al., 2014; Solazzo et al., 2017). A key goal of the studies and types of analysis above is to determine the

causes of biases between models and observations to enable improvements in modelling of ozone.





Typically spectral analysis allows the complex time series present in an ozone dataset to be decomposed into a set of spectral features. Studies have applied these methods to many parts of the world such as Europe, North America and Australia (e.g. Derwent et al., 2013, Young et al., 2013, Bowdalo et al., 2016), but to date no study has applied spectral analysis on global model and

observed ozone across India.

In this study, we have used Empirical Orthogonal Function (EOF) analysis on the OMI/MLS-observed and the model simulated TOC from HTAPII and CCMI.  EOF analysis reduces the dimensionality of the input spatial variables (i.e. ozone column, which is f(lat, lon, time)) to find new sets of variables that capture most of the observed variance from the original data through a linear

combination of the original variables. Principle Components (PC) represent the sign and overall amplitude of the EOF as a function of time. EOF analysis is commonly used in the climate science community (Nair et al., 2014), but has been less widely used in the ozone modelling community. EOF analysis is analogous to Fourier transform (FT) analysis, but performs better than FT when the signal differs from the pure sinusoidal waveform (Cepeda and Colome, 2014).

EOF analysis was applied to both the OMI/MLS and modelled TOC across a domain of $56^o$ to $105^o$ longitude and $5^o$ to $38^o$ latitude, which covers the entire Indian Subcontinent. Figure 12a depicts the spatial patterns of EOF1, which explains the maximum variance in tropospheric ozone over the domain. EOF1 has a loading for each variable, in this case the variables are the grid points, they have correlation structures both in space and time. The amplitudes of the EOF1 spatial patterns

have a time series as shown by PC1 in figure 12b.

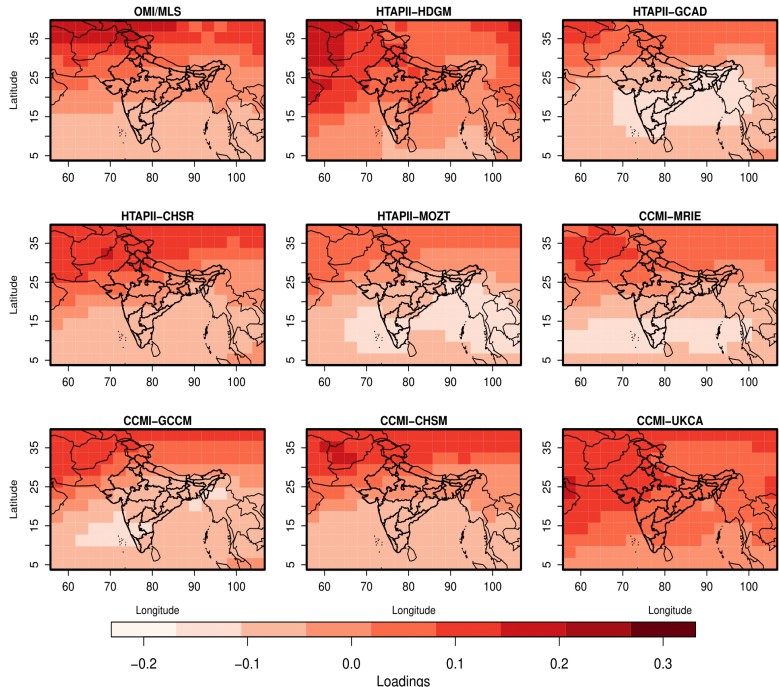




Figure 12: a) Dominant Spatial pattern (i.e. EOF1), which explains the maximum variance in the tropospheric ozone column.

5  The spatial patterns depicted by EOF1 (Fig12a) for models are similar to the spatial pattern for the OMI/MLS observations: they show higher values in the north-western part of domain and lower in the southern part and over the ocean. However, the magnitudes of the loading are different between each of the models and between the MMM and the observations.

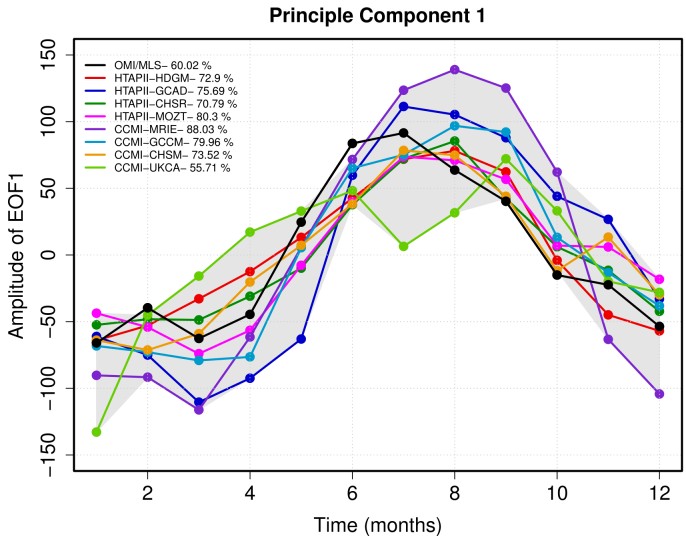

Figure 12:  b) Time series of the amplitude of EOF1 (PC1) with the values of accounted variance by the EOF1 in the legend for each model .

The amplitude of EOF1 (Fig. 12b) has negative values in winter and positive values during the
15  summer monsoon seasons. There is a discernible difference in the phase of PC1, with most of the models peaking in July-August, but the observations peaking in June. The annual-cycle like structure of PC1 shows a strong correlation with the movement of the ITCZ over India, which heads southward during winter and northward during summer. Physically these spatial patterns thus represent surface pressure changing with the movement of ITCZ. Precipitation also migrates with
20  ITCZ over India. Hence maximum variance in tropospheric ozone is explained by the monsoon over South Asia. It is worth noting that the maximum variance in tropospheric ozone column explained by EOF1 in observations is ~60% whereas in models it is greater than 70%. Maximum variance in tropospheric ozone column explained by EOF1 in CCMI-UKCA is ~55% which less than that of the observations. The differences in the EOF1 spatial pattern, the amplitudes of EOF1 (as given by the



PC1) and the percentage of maximum variance explained indicate that each model is capturing monsoon differently both in space and in time.

In spite of reasonable agreement between the models and observations for EOF1 and PC1, the comparison for EOF2 and PC2 is poor (Supplementary figures S2.a and S2.b). There is no
agreement between the spatial pattern of EOF2 and the amplitude of EOF2 (PC2) among the models and OMI-MLS. Whilst this EOF analysis has provided a novel approach to comparing and contrasting the modelled and observed tropospheric ozone column distributions, it does not give a clear understanding about the underlying reasons for the discrepancies in the models, as with many of the previous studies (Bowdalo et al., 2016; Derwent et al., 2013; Parrish et al., 2014; Solazzo et
al., 2017).

### 3.4 Comparison with the IAGOS-CARIBIC observations

We now focus on the comparison of the model data to vertical profiles of carbon monoxide and ozone measured on board a commercial airliner as part of the IAGOS-CARIBIC programme.
(Brenninkmeijer et al., 2007). The observations from IAGOS-CARIBIC are important as they provide a connection between the surface and satellite observations discussed above, but they are statistically less powerful owing to small samples sizes.

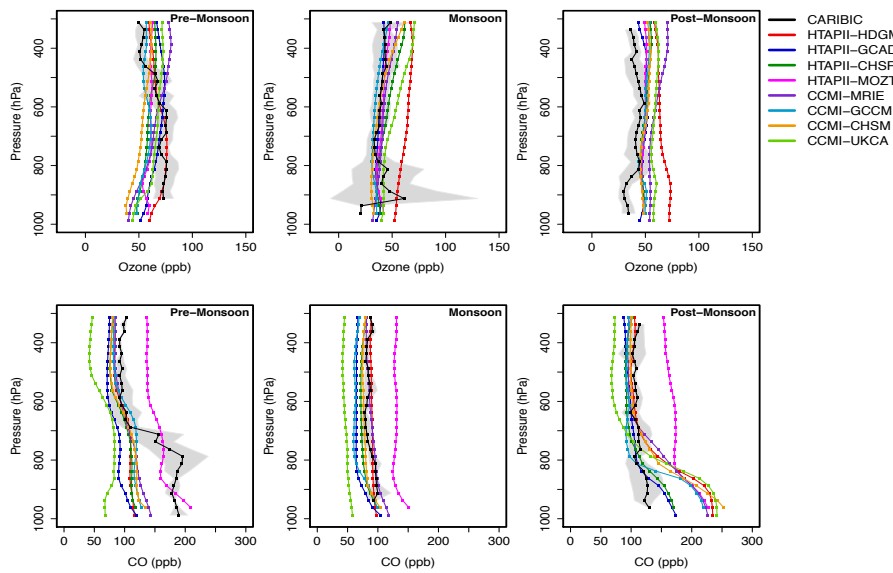

Figure 13: Comparison of ozone and carbon monoxide profiles from model simulations, 2010 with the CARIBIC
observations in Chennai for pre-monsoon (April-May), monsoon (June, July, August, September) and post-monsoon (October, November, December) seasons, 2008. Model simulations have been vertically interpolated along the CARIBIC flight pressure levels. Mean of the Data collected during the aircraft descent and ascent is shown here.



Figure 13 shows the seasonal mean vertical profiles of ozone and carbon monoxide from the IAGOS-CARIBIC aircraft observations from Lufthansa flights LH758 and LH759, which connect between Frankfurt, Germany, and Chennai, India, compared with model output over Chennai. In total we have combined the results from over 16 flights during April to December 2008. We have converted the IAGOS-CARIBIC data into pseudo-climatological data, by averaging over 25 hPa vertical bins as explained in Section 2.1.2, to enable a comparison of the models pre-, post- and during the monsoon. The black lines in Figure 13 denote the average observed vertical profile, with the grey envelope reflecting the standard deviation in these average observations. Model data (colours) refer to the average monthly means that coincide with the aircraft observations and are not samples along the aircraft flight tracks.

During the pre-monsoon season (April-May), high values of ozone and CO are observed in the Lower Troposphere (LT) ($p > 500$ hPa) as compared to the Upper Troposphere (UT) ($p < 500$ hPa). Generally speaking, models underestimate the ozone and CO values in the LT and perform fairly well in the UT. Given the fact that these are very limited observational data, any specific emission events (for example wild fires) that occurred during the observing period are unlikely to be reproduced by the models (Ojha et al., 2016). The levels of CO in the pre-monsoon LT are generally worse in comparison to the observations than the ozone levels. HTAPII-MOZT simulates the pre-monsoon LT carbon monoxide levels in good agreement with the observations, but highly overestimates the UT values and generally overestimates the CO mixing ratios in the post- and monsoon periods. CCMI-UKCA highly underestimates the CO profiles, especially in the UT. HTAPII-HDGM performs well in the LT for ozone profiles during the pre-monsoon season.

Chennai experienced a strong pollution event on the 15th of July 2010 (Ojha et al., 2016), hence high values of ozone are observed between 900-850 hPa during the monsoon season (June-July-August-September). Since the model ozone values are monthly mean values, models do not capture this strong pollution event. Aside from this event, models capture the ozone and CO profiles well during the monsoon season; the MMM bias is ~11% for ozone and ~ -5% for CO and correlation coefficient is ~0.29 for ozone and ~0.7 for CO. HTAPII-HDGM and CCMI-UKCA tend to overestimate the ozone profile in the UT whilst HTAPII-MOZT overestimates and CCMI-UKCA underestimates the CO profiles in the monsoon season.

There are large discrepancies between the models and IAGOS-CARIBIC observations in the LT during the post-monsoon season. Models overestimate the ozone and carbon monoxide profiles by a factor of 1.5 and 1.7, respectively, in the LT during the post-monsoon season (October-November-December). However, the models agree much better with the observed ozone and carbon monoxide profiles in the UT during this season. HTAPII-MOZT overestimates the carbon monoxide profile in the UT. The majority of the other models tend to have fairly high levels of carbon monoxide "trapped" within the boundary layer during the post-monsoon period. There is little



evidence for this trapping in the IAGOS-CARIBIC observations, but more evidence for pollutants (CO) build up in this season can be seen in the surface data analysed in Section 3.2.2.

The comparison of the HTAPII and CCMI models to these aircraft data have been useful in evaluating a basic evaluation of the vertical profiles of these key pollutants. However the limited

number of observed vertical profiles of these pollutants restrict detailed evaluation of models over this region. More, targeted aircraft based studies would be illuminating especially with comprehensive chemical and aerosol measurements to enable improvements in modelling in this region.

**3.5 Ozone as a function of VOC and NOx emissions**

Finally, in order to evaluate how the models are simulating ozone at the surface, we extend the analysis of surface ozone shown in Figure 3 to contrast the model simulated surface ozone against the model input VOC and NOx emissions following Squire et al. (2015) by creating ozone isopleth plots. Figure 14 shows the isopleths of surface ozone concentrations as a function of NOx and VOC

emissions for a subset of models (HTAPII-GCAD, HTAPII-CHSR, CCMI-GEOSCCM and CCMI-CHSM) over the entire domain of study. These models were chosen as they include (i) essentially the same model run for the two different MIPs (HTAPII-CHSR and CCMI-CHSM) (ii) different model runs for the same MIPs (HTAPII-CHSR and HTAPII-GCAD, CCMI-CHSM and CCMI-GEOSCCM) (iii) these were some of the only models that output total VOC emissions, which are better indicators

for ozone chemistry than carbon monoxide (Monks et al., 2015). The monthly mean surface ozone data over the study region from these simulations were combined with the monthly mean surface emissions of VOCs and NOx to generate the plots in Figure 14. The dots in each panel indicate the locations (in VOC and NOx space) that the model ozone data samples. As can be seen, there is wide variation in the VOC-NOx space sampled by the models due to differences in their input

emissions, as discussed in Section 2.2.





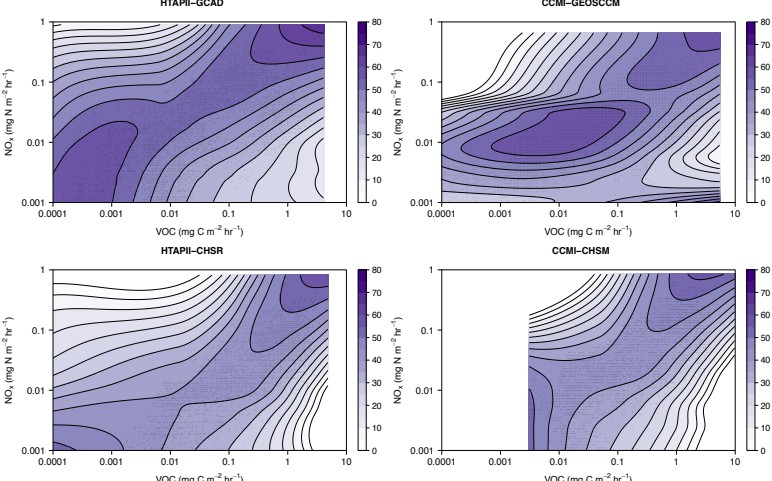

Figure 14: Isopleths of ozone concentration in ppb as a function of NOx and VOC emissions over the domain. The dots in each panel indicate the locations (in VOC and NOx space) that the model ozone data samples.

Unlike the ozone isopleths shown in Squire et al., 2015, which focused on grid boxes dominated by isoprene chemistry, the isopleths here generally show a double peak structure, with high ozone both at high and low NOx and VOC emissions (i.e. the bottom left and top right of each panel). This suggests that this analysis is not connecting in situ produced $O_3$ to the underlying emissions of VOCs and NOx and shows effects of pollutants from other regions as well. HTAPII-CHSR and

CCMI-CHSM have the same chemistry scheme but the different inputs used cause the ozone to respond differently to the NOx and VOC emissions. The HTAPII models (HTAPII-CHSR and HTAPII-GCAD) have the same anthropogenic emission inputs but the difference in the chemistry scheme used causes the ozone to respond differently to the NOx and VOC emissions. On similar lines, the CCMI models (CCMI-CHSM and CCMI-GEOSCCM) also give different isopleth patterns.

### 4. Conclusions

In this study, we have systematically assessed differences and similarities in the modeled ozone from eight different models, contributing to the HTAPII and CCMI model inter-comparison projects, over the Indian sub-continent. Large inter-model variability is observed in the model simulated

annual average surface. Tropospheric $O_3$ and ozone precursors from these models have been evaluated against a set of ground based, aircraft and satellite observations over India. Comparison between the model simulated and ground based observations of surface ozone show some similarities between the seasonal cycle, except at Chennai. However, models overestimate the ozone mixing ratios at all locations, with CCMI-UKCA giving the highest values of annual average

surface ozone.




While a detailed evaluation of why CCMI-UKCA simulates the highest levels of annual mean surface ozone is beyond the scope of this study, we note that further work should be performed to understand the reasons behind this behaviour. Simulations similar to those in Prather et al., 2018 would potentially help shed light on the role of the chemical scheme as a source of bias in the

model.

Models underestimate NOx mixing ratios, except HTAPII-HDGM at Patiala. $NO_2$ dominates NOx in the models. Models tend to underestimate CO only at Delhi and Patiala and perform well at the other ground based stations. It is important to note that the sites considered in this study are categorised as semi-urban and are therefore influenced by local emissions, which are not well

represented in global models. Coarse grid resolution models are unable to capture the short time scale processes taking place at the local scale and result in the underestimation of surface carbon monoxide and NOx and the overestimation of ozone as we have shown in Figures 3-7. In order to better evaluate global model simulations of surface ozone, we would suggest the need for a network of rural stations measuring ozone and ozone precursors (i.e. NOx, CO, VOCs), covering different

geographical and chemical environment across India.

Model simulations of total TOC show similar spatial patterns compared to the OMI data over the study domain, but they overestimate the total TOC values with biases ranging from 16% to 40%. EOF analysis highlights that more than 70% of the ozone variation in models is dependent on a single phenomena i.e. EOF1.

Comparison with the CARIBIC ozone and CO profiles indicate that models perform fairly well in the upper troposphere as compared to the lower troposphere. The sparse observations of CO and $O_3$ profiles limit the evaluation of model ozone and CO profiles over this region. It is clear from the ozone isopleths that different inputs and chemistry schemes used in these models cause the ozone to respond differently to VOCs and NOx emissions. Large variation in lightning NOx emissions is

one of the major reason for the differences in the total NOx emissions. Further investigation to support this study including the details of chemistry schemes and the simulations of VOC, $HO_2$ needs to be evaluated within each model. For future chemistry-climate model intercomparisons, we recommend inclusion of simulations with standardisation of non-anthropogenic emission sources as well as anthropogenic sources in order to be able to diagnose the impact of model chemistry only on

tropospheric ozone.

*Acknowledgements.* The author acknowledges the ISAAC-Newton trust and NERC APHH project, PROMOTE (NE/P016383/1) for funding. We thank the System for Air Quality and Weather Forecasting and Research (SAFAR) project and the Modelling Atmospheric Pollution and
Networking (MAPAN) project, Indian Institute of Tropical Meteorology, Pune, India, for the ground based measurements used in this study. Gerd A. Folberth was supported by the Joint UK BEIS/Defra Met Office Hadley Centre Climate Programme (GA01101) and the European Union's Horizon 2020 Research and Innovation Programme under grant agreement no. 641816 (CRESCENDO). GEOSCCM is supported by the NASA MAP program and the high-performance computing
resources were provided by the NASA Centre for Climate Simulation (NCCS). We thank S.M. Frith



for helping us access the GEOSCCM output. We also thank Makoto Deushi, Meterological Research Institute (MRI), Japan for helping us access the MRI-ESM1r1 simulations and Yanko Davila, Department of Mechanical Engineering, University of Colorado, Boulder, CO, USA, for the GEOSCHEM-ADJOINT simulations for the present study. UMUKCA-UCAM model integrations were performed using the ARCHER UK National Supercomputing Service and MONSooN system, a collaborative facility supplied under the Joint Weather and Climate Research Programme, which is a strategic partnership between the UK Met Office and the Natural Environment Research Council. We thank CARIBIC partners as well as Lufthansa, especially Lufthansa Technik for support. We acknowledge the AURA MLS and OMI instruments and algorithm teams for the satellite measurements used in this study.



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
