# Peer review of "Evaluation of tropospheric ozone and ozone precursors in simulations from the HTAPII and CCMI model intercomparisons - a focus on the Indian Subcontinent."

_Atmospheric Chemistry and Physics, 2018_

## Referee Comment (RC1) · Ojha (Referee) · 28 Dec 2018

O$_3$ pollution over Indian subcontinent causes considerable losses in the crop productivity and affects human health leading also to pre-mature mortalities. Considering scarcity of in situ measurements in the region, manuscript by Hakim et al. presenting the comparison of ozone simulations among several models and with available observations is of great interest. Manuscript is recommended for publication in the Atmospheric Chemistry and Physics. Following comments and suggestions should be considered during the revision.

Comparison of model results are primarily made with urban / semi-urban environments. I agree with authors (Page 1, l:31–33 and Page 28, l: 8–10) that global models, due to coarse resolution could have limitation in reproducing local influences. It might be useful to also compare with recent observations considerably away from major anthropogenic influences, such as Nainital (Sarangi et al., 2014).

Why Delhi site is considered as semi-urban? where average values of NOx up to 180 ppbv (Page:17, l:28–29) indicate strong anthropogenic influences. It is possibly better to classify this site as urban. In addition, in the text, a mention of O$_3$, CO, NOx observational values should also be mentioned from other stations in Delhi (e.g. Sharma et al., 2016) (which would be within a grid box of global models). This would provide a more general range of NOx bias over Delhi, which seems as of now very high (based on values at single location in the region of strongest variability).

Figure 4 and Page 13, l.10–13: This analysis is very useful and tells clearly over which regions models differed with each other, more strongly. The text "This is worse than…….Europe and North America" should include quantitative information on what are typical % standard deviations (or range) seen in MIPs over Europe and North America, for a ready reference here itself.

Page 14, l.9–10: "In all locations….than observations". No, Jabalpur mean model values seem well within 1-sigma of the observed values. Check and modify the statement suitably.

Page 19, l.11–16: Pl. check for a consistency in this text. L.13 says that at Jabalpur correlation is poor, then it is said that "models show good correlation at all sites" in l.15.

Page 20, l.8–9: High levels of tropospheric ozone columns (TOC) are attributed to anthropogenic activities and biomass burning. While for surface ozone it could be the driver, tropospheric column ozone could have considerable contribution from long-range transport and Stratosphere-to-Troposphere Transport (STT) (see e. g. Ojha et al., 2017). I do not see any mention of these aspects here. Is it possible to further compare whether stratospheric contributions among models are similar to each other or they differ significantly? This could corroborate the finding (Page 21, l.2–4) that the CCMI-UKCA produces highest surface ozone but not the TOC, indicating potential influence of processes (other than regional emissions) affecting the inter-comparison of model TOC.

Page 24, Figure 13: Model simulations were said to be for period 2008–2010 (Abstract: Page 1, l.21). For comparison with CARBIC observations during 2008, why model data for 2010 is used (and not for same year 2008). Did I miss something here?

Minor comments:
Page 7, l. 22: delete "traditionally"; and consider changing "observation poor" to "observationally sparse"

Page 10, l.1–4: The sentence should be reframed.
Page 16, l.23 and Page 17, l.13: "between models" to "among models"
Page 25, l.9–10: This is not clear. Do you mean that model profiles over the Chennai airport are used for comparison? If yes then write so.
Page 25, l.16–17: "The levels of CO ..are generally worse in comparison…". consider rewording the sentence.
Page 25, l.22: Correct "2010" to "2008"

Sincerely,

Narendra Ojha
Physical Research Laboratory, India

References

Ojha, N., Pozzer, A., Akritidis, D., and Lelieveld, J. (2017), Secondary ozone peaks in the troposphere over the Himalayas; *Atmos. Chem. Phys.*, 17, 6743-6757, https://doi.org/10.5194/acp-17-6743-2017.

Sarangi, T., Naja, M., Ojha, N., Kumar, R., Lal, S., Venkataramani, S., Kumar, A., Sagar, R., and Chandola, H. C. (2014), First simultaneous measurements of ozone, CO, and NOy at a high-altitude regional representative site in the central Himalayas, *J. Geophys. Res. Atmos.*, 119, 1592–1611, doi: 10.1002/2013JD020631.

Sharma, A., Sharma, A. K., Rohtash, Mandal, T. K. (2016), Influence of ozone precursors and particulate matter on the variation of surface ozone at an urban site of Delhi, India, *Sustain. Environ. Res.*, http://dx.doi.org/10.1016/j.serj.2015.10.001.

---

## Referee Comment (RC2) · Anonymous Referee #2 · 15 Feb 2019

Hakim et al., present results for intercomparison of HTAPII and CCMI models on the evaluation of ozone levels over India sing different observations, where surface observations are scarce. It is an interesting study and identifies some key challenges of climate models in reproducing observed ozone levels over india. I have a number of questions and comments in addition to reviewer #1 before the manuscript can be accepted for publication in ACP.

1) Are there no rural sites to be used in model evaluation?

2 ) Figure 1 caption: Please make clear that these are total (anthropogenic + natural emissions).

3) Line colors in Figure 2 are difficult to be attributed to the individual models, please consider changing color scale.

4) Figure 3. Change "Mean" to "MMM" in the figure to be consistent with the text.

5) In Table 1 caption, also refer to Fig. 4 for monitoring site locations.

6 ) Are there any filtering for missing data in the calculation of monthly mean observations?

7) What is special about Chennai that leads to poor temporal model evaluation?

8) Among the monitoring sites, Delhi seems to have much higher NOx and CO values compared to the other sites. Values reaching to almost 200 ppb for NOx and 5 ppm for CO do suggest that this station is not a typical sub-urban station. Can the authors comment on this?

9) It is interesting that for CO, the two stations with poorest correlations have the lowest biases judged from Fig. 9. Can the authors comment on this?

10) What is the difference between AATOC and MTOC?

11) What do the PC1 and PC2 components refer to in these analyses? Is PC1 the monsoon system?

12) Can the reason why EOF do not provide clear understanding in comparison to other studies be the temporal resolution of o3? Can i.e. looking at daily or sub daily resolutions give more answers (like in i.e. Solazzo et al., 2017)

13) Adding MMM in Figure 13. could be helpful to interpret results.

---

## Author Comment (AC1) · 26 Mar 2019

Response to reviews of "Evaluation of tropospheric ozone and ozone precursors in simulations from the HTAPII and CCMI model intercomparisons - a focus on the Indian Subcontinent".

We would like to thank the editor and the two reviewers for their time in handling and reviewing our manuscript. We reply below to each of the reviewers comments in turn.

Reviewer 1
*O3 pollution over Indian subcontinent causes considerable losses in the crop productivity and affects human health leading also to pre-mature mortalities. Considering scarcity of in situ measurements in the region, manuscript by Hakim et al. presenting the comparison of ozone simulations among several models and with available observations is of great interest.*

*Comment 1: Manuscript is recommended for publication in the Atmospheric Chemistry and Physics. Following comments and suggestions should be considered during the revision. Comparison of model results are primarily made with urban / semi-urban environments. I agree with authors (Page 1, l:31–33 and Page 28, l: 8–10) that global models, due to coarse resolution could have limitation in reproducing local influences. It might be useful to also compare with recent observations considerably away from major anthropogenic influences, such as Nainital (Sarangi et al., 2014).*

Firstly, we would like to thank Prof. Ojha for his time in reviewing our manuscript. We agree that this paper should be of wide interest as we evaluate the output from multiple models with the widest array of observations of $O_3$ collected to-date. The ground based observations used in this study were obtained from a network of monitoring stations under the MAPAN project (run by the Indian Institute of Tropical Meteorology, Pune, India). Each station is designed to be as similar as possible in its micro-environment (e.g. placement relative to large obstructions etc.) and the sites are located away from road sides in more open areas – albeit in urban regions. Of the sites we investigated all but one (Lodhi Road, Delhi) can be classified as semi-urban (similar to the UK London Ealtham site, part of the UK DEFRA network). The Lodhi Road site itself is classified as an urban-background site. Thus these MAPAN sites are not measuring at the direct emission sources but are reflective of the wider urban atmosphere. For some regions this will inevitably be quite heterogenous but for larger places, such as Delhi, the levels of NOx, CO and subsequently O3 are relatively similar across the region. Clearly global models will struggle to reproduce the road side concentrations next to emission sources but we argue that the wider scale features of the urban composition should be reproducible by these models and that the comparison against these data is instructive. Rural observations are very scarce across the world and particularly in India (largely given that the monitoring focus is around human exposure to pollution and compliance). We have been in contact with the authors of the Sarangi study and would have liked to have included a comparison against the data from Nainital but to this date we have not had any reply and so have omitted it from our analysis. As we recommend in the abstract "a higher density of long term monitoring sites measuring not only ozone but also ozone precursors including speciated VOCs, located in more rural regions of the

Indian sub-continent, would enable improvements in assessing the biases in models run at the resolution found in HTAPII and CCMI". We hope that in the future these measurements are made and that follow up studies can assess them.

*Comment 2: Why Delhi site is considered as semi-urban? where average values of NOx up to 180 ppbv (Page:17, l:28–29) indicate strong anthropogenic influences. It is possibly better to classify this site as urban. In addition, in the text, a mention of O$_3$, CO, NOx observational values should also be mentioned from other stations in Delhi (e.g. Sharma et al., 2016) (which would be within a grid box of global models). This would provide a more general range of NOx bias over Delhi, which seems as of now very high (based on values at single location in the region of strongest variability).*

We agree with Prof. Ojah that the Lodhi Road station in Delhi has very high NOx levels and its classification as a semi-urban site was wrong. We clarify this by amending the classification as an urban-background site in the text. Interestingly the Lodhi road is inside a relatively green area and is well away from the road-side pointing to very widespread issues with NOx pollution in Delhi. More detailed work looking at the atmospheric composition across Delhi is planned for the future.

*Comment 3: Figure 4 and Page 13, l.10–13: This analysis is very useful and tells clearly over which regions models differed with each other, more strongly. The text "This is worse than.......Europe and North America" should include quantitative information on what are typical % standard deviations (or range) seen in MIPs over Europe and North America, for a ready reference here itself.*

The section of the text Prof. Ojha is referring to compares the relative variability in the models surface O3 we have looked at across the Indian subcontinent (Figure 4) with the variability shown by Young et al (2013) for the ACCMIP models. The statement we made was overly negative and we have revised this in the main text.

We have also added in as the referee suggest some text to explain based on Young et al (2013) what typical relative variability in the models surface O3 there is in the more studied regions of North America and Europe.

The revised text now reads: "The standard deviation of the multi model ensemble is shown in Figure 4. The standard deviation of the multi model mean can be used as an indicator of the level of agreement between the models. Here we show that there is a reasonably low level of agreement between the models, with an average of 23% standard deviation in the mean. This is slightly worse than the level of agreement between the ACCMIP models over the same region shown in Young et al., 2013 (< 20% standard deviation in the mean) and could reflect the fact that here we compare simulations from two different MIPs which make use of different emissions. However, we find the difference between the emissions within models of a particular MIP is as large as those between MIPs (Figures 1, S2 and S3). Figure 4 highlights that models differ most in the northern and eastern part of India and

standard deviation is the least in the central part of India. For the more well studied regions such as North America and Europe, Young et al. (2013) show that global model multi model analyses have similar if not slightly larger variability than over the Indian sub-continent. Young et al. (2013) show that the variability in the South East USA is very high, > 30%, across the ACCMIP models, which is likely linked to the impacts of different biogenic emissions (not specified in MIP protocols) and chemistry over this isoprene rich area."

*Comment 4: Page 14, l.9–10: "In all locations....than observations". No, Jabalpur mean model values seem well within 1-sigma of the observed values. Check and modify the statement suitably.*

We thank the reviewer for highlighting this and have modified the manuscript accordingly. The text now reads "In seven out of eight, the ozone mixing ratio is higher in the MMM than in the observations (except Jabalpur, where MMM is within 1-sigma deviation)."

*Comment 5: Page 19, l.11–16: Pl. check for a consistency in this text. L.13 says that at Jabalpur correlation is poor, then it is said that "models show good correlation at all sites" in l.15.*

Again, we thank Prof. Ojah and correct the text to read: "The model simulations capture the seasonal variability in monthly mean CO well (R-values > 0.4 for all models) at most locations; the exception is in Hyderabad where all models generally show a negative correlation with the observations and at Jabalpur where correlation is poor (see section S5 of supplementary material). Interestingly, the model simulations at Jabalpur and Hyderabad show lowest correlations with the observations in spite of having the lowest biases. This could point towards some important processes which the models are struggling to simulate but further work would be needed to clarify this. The site with the best correlation is Udaipur, where the MMM correlation coefficient is 0.96. Models are in agreement with the observed CO at all sites but highly underestimate the observed values at Delhi and Patiala."

*Comment 6: Page 20, l.8–9: High levels of tropospheric ozone columns (TOC) are attributed to anthropogenic activities and biomass burning. While for surface ozone it could be the driver, tropospheric column ozone could have considerable contribution from long-range transport and Stratosphere-to-Troposphere Transport (STT) (see e.g. Ojha et al., 2017). I do not see any mention of these aspects here. Is it possible to further compare whether stratospheric contributions among models are similar to each other or they differ significantly? This could corroborate the finding (Page 21, l.2–4) that the CCMI-UKCA produces highest surface ozone but not the TOC, indicating potential influence of processes (other than regional emissions) affecting the inter-comparison of model TOC.*

We agree with Prof. Ojah that this is an interesting area to analyse further and identify the role of strat-trop-transport of $O_3$ (STT-$O_3$) across the region. Indeed, in his work he has shown that this is an important processes in some regions of the

domain we have assessed. However, to do this systematically requires that all models have a diagnostic of the STT-$O_3$ (sometimes called the O3S tracer) and this was not available for the models we've looked at. However, a follow up study would nicely compare the bias in the simulated $O_3$ with this tracer.

*Comment 7: Page 24, Figure 13: Model simulations were said to be for period 2008–2010 (Abstract: Page 1, l.21). For comparison with CARBIC observations during 2008, why model data for 2010 is used (and not for same year 2008). Did I miss something here?*

We are sorry about the confusion here. We've only used model data for one year (2009 for HTAPII-HADGM 2010 for all other models). We will change the abstract and tidy the text to make the point clearer. We were unable to find any CARIBIC data for 2010 and instead used the most recent data we could find which is for 2008. Comparing the model output for 2009 and 2010 against CARIBIC data for 2008 doesn't change the picture and we wanted to stick with a consistent base model year (2010).

Minor Comments:

*Page 7, l. 22: delete "traditionally"; and consider changing "observation poor" to "observationally sparse"*

Minor comment 1 – Done.

*Page 10, l.1–4: The sentence should be reframed.*

Minor comment 2 – Done.

Page 16, l.23 and Page 17, l.13: "between models" to "among models"

Minor comment 3 – Changed as requested.

Page 25, l.9–10: This is not clear. Do you mean that model profiles over the Chennai airport are used for comparison? If yes then write so

Minor comment 4 – Thanks for the comment, we have changed the text to clarify this. The text now reads "Model data refer to the average monthly mean model profiles over Chennai airport that coincide with aircraft observations, also interpolated to 25hPa vertical pressure bins."

Page 25, l.16–17: "The levels of CO ..are generally worse in comparison…". consider rewording the sentence.

Minor comment 5 – Done. The corrected text now reads "The levels of model simulated CO in the pre-monsoon LT generally show higher biases as compared to the ozone levels. HTAPII-MOZT simulates the pre-monsoon LT carbon monoxide

levels in good agreement with the observations, but highly overestimates the UT values and generally overestimates the CO mixing ratios in the post- and monsoon periods."

Page 25, l.22: Correct "2010" to "2008"

Minor comment 6 – Corrected accordingly.

---

## Author Comment (AC2) · 26 Mar 2019

Response to reviews of "Evaluation of tropospheric ozone and ozone precursors in simulations from the HTAPII and CCMI model intercomparisons - a focus on the Indian Subcontinent".

We would like to thank the editor and the two reviewers for their time in handling and reviewing our manuscript. We reply below to each of the reviewers comments in turn.

Reviewer 2:

*Hakim et al., present results for intercomparison of HTAPII and CCMI models on the evaluation of ozone levels over India sing different observations, where surface observations are scarce. It is an interesting study and identifies some key challenges of climate models in reproducing observed ozone levels over india. I have a number of questions and comments in addition to reviewer #1 before the manuscript can be accepted for publication in ACP.*

We thank anonymous reviewer 2 for their time and insightful comments to help improve our paper. We reply to these comments below.

*1) Are there no rural sites to be used in model evaluation?*
We have used a systematic set of surface observations from the MAPAN network. Whilst there are a limited number of rural sites across the domain (see comment from reviewer 1) these are generally located in regions of very clean air (i.e. in the Himalaya region) or are not available to us. We are not sure if the reviewer feels that rural sites would improve the evaluation, we infer they do, but we think that this is key and the next step forward to understanding how to improve model representation of this area. See response to reviewer 1 above and the modified manuscript for further details.

*2) Figure 1 caption: Please make clear that these are total (anthropogenic + natural emissions).*
We have modified the Figure caption accordingly.

*3) Line colors in Figure 2 are difficult to be attributed to the individual models, please consider changing color scale.*
We thank the reviewer for pointing this out and have changed the line colours in Figure 2 for consistency with the other figures.

*4) Figure 3. Change "Mean" to "MMM" in the figure to be consistent with the text*
We have changed this to be consistent.

*5) In Table 1 caption, also refer to Fig. 4 for monitoring site locations.*
We have modified Table 1 to make the links clearer.

6) Are there any filtering for missing data in the calculation of monthly mean observations?

Inevitably there are always gaps in any data record. In our analysis we considered missing values to be "NA" values and these were omitted while taking the mean. The data gaps are pretty small and we have calculated these here for the reviewer. The percentage of days missing per station across the year analysed were:

Delhi – 5.4% (20 days)
Patiala – 8.4% (31 days)
Udaipur – 0% (0 days)

Jabalpur – 1.9% (7 days)
Pune – 3.8% (14 days)
Guwahati – 1.9% (7 days)
Chennai - 0% (0 days)

7) What is special about Chennai that leads to poor temporal model evaluation?

The model the grid box(es) that we used to evaluate the performance at Chennai were heavily influenced by having large ocean fractions. This was the only station we looked at with the models that was heavily influenced by the coast and coupled to that   Chennai is affected by both summer and winter monsoons. This combination is likely responsible for the poor model temporal evolution and we have modified the text to make the case for further work looking at coastal sites in the region. The text now reads "Observations at Chennai peak in April and October, i.e. during pre and post summer monsoon season. Models show poor correlation with the seasonal cycle of ozone at Chennai. To some extent this might be affected by the model's ability to simulate summer monsoon (from the south-west) and winter monsoons (from the north-east) that affect Chennai. It would be worth comparing model simulations with ozone observational data at Mumbai on the west coast of India, which receives rainfall only during the summer to understand the role of the monsoon near these coastal sites and we suggest further analysis assesses the performance of the models at the coastal impacted locations specifically."

8) Among the monitoring sites, Delhi seems to have much higher NOx and CO values compared to the other sites. Values reaching to almost 200 ppb for NOx and 5 ppm for CO do suggest that this station is not a typical sub-urban station. Can the authors comment on this?

Please see our reply to Reviewer 1 Comment 2.

9) It is interesting that for CO, the two stations with poorest correlations have the lowest biases judged from Fig. 9. Can the authors comment on this?

We agree with the reviewer that it is interesting but also note that there is no reason why correlation should be related to bias. It is interesting and we have to consider whether or not it is related to the parameterised processes like chemistry (which arguably are more likely to be related to the correlation coefficients) or inputs (i.e.

emissions). We can't say at this stage which is cause and which is effect but we agree with the reviewer and note the interesting feature in the main paper. We have added the following text to the paper to make the point "Interestingly, the model simulations at Jabalpur and Hyderabad show lowest correlations with the observations in spite of having the lowest biases. This could point towards some important processes which the models are struggling to simulate but further work would be needed to clarify this."

10) What is the difference between AATOC and MTOC?
AATOC is annual average of total ozone column and MTOC is the spatial mean of AATOC over the region considered in this study. We have modified the text to make this clearer "In order to evaluate the model simulations and observations we first compare the mean total ozone column (MTOC), defined as the spatial mean of AATOC over the study domain."

*11) What do the PC1 and PC2 components refer to in these analyses? Is PC1 the monsoon system?*
The reviewer is correct in suggesting that the timing of PC1 reflects the monsoon system. We have made changes in the paper to make this clearer "Hence maximum variance in tropospheric ozone is explained by the monsoon over South Asia (i.e. PC1 reflects the monsoon)." However, PC2 is more complex and at present we don't have a strong physical argument for what it represents. More work is required to tackle this which is beyond the scope of the present manuscript.

12) Can the reason why EOF do not provide clear understanding in comparison to other studies be the temporal resolution of o3? Can i.e. looking at daily or sub daily resolutions give more answers (like in i.e. Solazzo et al., 2017)
We very much like the Solazzo et al (2017) study and would like to follow that approach up in further studies looking at the ground based observations if we are able to get sub hourly data (at present hourly data is the highest time frequency). The satellite data we analysed to generate the EOFs are monthly means. The orbiting of the satellite means that at present daily data is the highest time frequency available and we could look at daily slices but there are issues with sampling, due to clouds etc, which mean that the analysis is much more robust by looking at monthly mean data. However, if and when higher time resolution satellite data becomes available it would be incredibly useful to perform new analyses with these data to better understand the temporal dependence of $O_3$ both in models and reality.

13) Adding MMM in Figure 13. could be helpful to interpret results
We have modified the Figure 13 accordingly.